# TEEDA: An Interactive Platform for Matching Data Providers and Users in the Data Marketplace

**Teruaki Hayashi * and Yukio Ohsawa**

Department of Systems Innovation, School of Engineering, The University of Tokyo, Tokyo 113-8656, Japan; ohsawa@sys.t.u-tokyo.ac.jp
* Correspondence: hayashi@sys.t.u-tokyo.ac.jp

**Abstract:** Improvements in Web platforms for data exchange and trading are creating more opportunities for users to obtain data from data providers of different domains. However, the current data exchange platforms are limited to unilateral information provision from data providers to users. In contrast, there are insufficient means for data providers to learn what kinds of data users desire and for what purposes. In this paper, we propose and discuss the description items for sharing users' calls for data as data requests in the data marketplace. We also discuss structural differences in data requests and providable data using variables, as well as possibilities of data matching. In the study, we developed an interactive platform, named "treasuring every encounter of data affairs" (TEEDA), to facilitate matching and interactions between data providers and users. The basic features of TEEDA are described in this paper. From experiments, we found the same distributions of the frequency of variables but different distributions of the number of variables in each piece of data, which are important factors to consider in the discussion of data matching in the data marketplace.

**Keywords:** matching; data marketplace; data platform; data visualization; call for data

## 1. Introduction

Big data and artificial intelligence have remained a global trend [1,2], and several types of data that existing analytical technologies cannot handle have been frequently encountered. Social movements have also increased attention toward areas not centered on these technologies. Rather than relying on a single data source, expectations have arisen in solving problems and finding new values in data through the distribution, exchange, and linking of data across various fields [3–5]. Thus, data has become a transferrable and interchangeable resource in the digital economy [6,7].

Recently, various forms of data marketplaces have been launched as platforms [5,7–11]. Moreover, linked data [12], ontology matching [13], and the development of data standardization and data catalogs such as data catalog vocabulary (https://www.w3.org/TR/vocab-dcat/) and semantic sensor network ontology (https://www.w3.org/TR/vocab-ssn/) have enabled searches across databases that were closed to each domain. These services and technologies have made it easier to link and exchange data from different fields and have functions of a collaborative environment related to data, such as the discovery of related knowledge by structured knowledge for data utilization [14], a communication environment that guarantees reliability by consortia [9,15], and an integrated environment for data analysis [16,17].

Although there are numerous data exchange services and platforms on the Web, they only provide unilateral information from the data providers. Most of the technologies, such as data catalogs and data merging methods, are for supporting data providers in making their data available in the marketplace. However, the means for data providers to learn what kinds of data users (the data buyers) desire and for what purpose are insufficient. In contrast, methods for data users to request data have been

inadequately discussed. Therefore, the distribution and trading of valuable data has been hindered. To solve this problem, the following two approaches are necessary in the data marketplace:

1. Description items of the users' calls for data as data requests;
2. A platform where the data information (users' data requests and providable data of data providers) converges.

In this paper, we propose the description items of data requests and present a matching platform, named "treasuring every encounter of data affairs" (TEEDA), for externalizing, sharing, and matching data requests with providable data in response to the above issues. Data requests, in this paper, are the needs of data users, and providable data are the information on data that can be provided to the marketplace by data providers. TEEDA facilitates connection between data providers and users who seek data to suit their needs; thus, data providers can learn the needs of users and provide suitable sets of data. In this study, in addition to describing the basic features of the matching platform, we analyzed two types of data information collected by TEEDA. Subsequently, we discuss their structural differences using variables, as well as the possibilities of data matching.

Previous papers have reported research on data marketplaces and data exchange platforms where data are traded as exchangeable economic goods [18,19]. These studies have explored game theory [20], privacy models [21,22], a market model for innovative collaboration [5], a secure model using blockchain [10], pricing mechanisms of data [23,24], and the complex network approach to conceptualize data exchange platforms [25]. Therefore, considering data matching in the data marketplace is a natural extension of these previous works. Furthermore, data platforms to encourage cross-disciplinary data-driven innovation have been proposed in the literature; examples of these platforms include DataHub [16], Labbook [17], and the DJ site (https://datajacket.org/?lang=english) or DJ store [14]. These platforms not only function as data portal sites, but also enable the discovery of data and sharing of analytical knowledge between various stakeholders in academia and industry. However, as discussed earlier, these platforms primarily provide unilateral information on data by the data providers themselves, which is insufficiently compatible with users' calls for data. In contrast, Virtuora DX [9], D-Ocean [11], or Web-based IMDJ [15] have functionalities that enable the sharing of users' opinions via chat. However, data provided in response to these calls are in the form of free text, which does not define the description items necessary to appropriately express data attributes including variables. Moreover, data matching to achieve matching between different types of data sources, such as schema matching or record linkage, has also been previously discussed [13,26]. However, the primary focus of the current study is to not only explore the linkages between data themselves, but also match the data providers and users via metadata. Thus, the novelty of this study is not only proposing a description framework of users' needs and providing a platform with functions to convey these needs to data providers, but also exploring data matching approaches.

Matching problems were first addressed via Gale and Shapley's stable matching [27], which is famously referred to as the marriage problem and has been since developed in various ways. In the data marketplace, considering a market where stakeholders are matched through data can be considered a natural extension of market design. As mentioned above, the data market is an emerging market, and to the best of our knowledge, the current study is the first to explore data matching in the data marketplace. Therefore, we made several assumptions in order to understand and discuss data matching in the data market. First, in addition to data providers and users, there are several other stakeholders, such as data brokers and analysts [19,28–30]. Although there is a matching problem considering two or more players [31], for the sake of simplicity, we only considered the matching between the data providers and users. Second, it has been noted that data users do not always have sufficient knowledge about data, which makes it difficult to express the data [14]. However, in this study, to simplify the model, we assumed that data users can sufficiently describe the data they want in the form of data requests. Third, for data matching, we considered a data-specific feature: duplicability. In the marriage problem, every participant can match at most one person, and this case

is called one-to-one matching. In contrast, there is many-to-one matching for firms and workers [32]. As data can be duplicated easily, we must consider many-to-many matching. In this paper, we discuss the matching possibility using the similarity of data structures based on common variables without using the concept of preference. The most important contribution of TEEDA is to make it possible to externalize the data users' calls for data, which has not been addressed in previous studies and platform services.

The remaining part of this paper is organized as follows. In Section 2, we explain the design of TEEDA based on the description items of data requests and providable data, and we demonstrate the functions of the matching platform. In Section 3, we present the experimental details of this study. In Section 4, we discuss the results obtained from our experiment and mention areas of future work. Finally, we provide concluding remarks in Section 5.

## 2. Design and Implementation

An important feature of TEEDA is the ability to allow both data providers and users to share information on data and investigate the possibility of matching with each other. By calculating and visualizing detailed matching, data providers can learn what kinds of data are required and for what purposes. Similarly, users can understand what kinds of data are available and who owns the data.

### 2.1. Description Items to Share Data Requests

The emergence of platforms for data marketplaces has made it possible to obtain information about various data from different domains. However, although there are data on the premise of disclosure, such as open data, data are not always open to the public. It is difficult to share data when the usage is not clear. Therefore, information on data, such as their whereabouts or outlines, is provided as metadata. Standard methods for describing metadata (data catalogs), such as the Dublin core schema or data catalog vocabulary, have been proposed for data providers to express a summary of their own data. Metadata description makes it possible to share information on data with others while reducing privacy risks and the risk of loss of business opportunities in the data marketplace.

The above methods for metadata description, however, implicitly assume that the data users are familiar with the data. That is, users who read the metadata must possess sufficient knowledge about the structure and the contents of the data. However, the data marketplace discussed in this paper is a place where various stakeholders with different background knowledge discover and exchange data. Therefore, the data users do not always have specialized knowledge about the data. To match data providers and users, the data providers must provide information on the data in a concise and understandable manner for the users. Similarly, data users must be able to explain and describe a summary of the data they want. Furthermore, both descriptions of the data must be in a format that is human friendly to easily understand the contents and machine readable to easily calculate the relationships. Methods for metadata description that allow users to express the type of data with the kinds of variables they want have not been established.

The description items of the data requests and providable data proposed in this paper are shown in Table 1. We use metadata description with the assumption that the core attributes of the data are data name and a set of variables. The data name is an item to express the data that the users want in natural language. We can learn and specify the areas and types of required data, e.g., "earthquake data in the great east Japan earthquake" or "tweets during the Christmas season in 2018." A variable is a logical set of data attributes [33] that are features important to understanding the structure of the data and determines the granularity of the data [25,34]. In "earthquake data of the great east Japan earthquake", for example, "year", "month", "day", and "time" are necessary to learn when the earthquake occurred; the "latitude", "longitude", "depth", and "center name of the earthquake" are to learn where the earthquake occurred; and "magnitude" is to learn the strength of the earthquake. Hence, we can understand that the user wants to learn the seismic magnitude scale of an earthquake in an area. Therefore, a set of variables is an attribute important to learning about the data requests from

users. Further, as supplementary information, we introduced the item "purpose of data use" to enable data providers to understand the structure of data that users want. In this study, the purpose of data use is not an essential item in the data request as data users do not always want to initially reveal their intent in obtaining data due to the potential loss of business opportunities.

**Table 1.** Description items of data requests and providable data (* denotes a required item).

| Type | Field Name | Description |
|---|---|---|
| Data request | Data name * | The name of requested data |
| | Variables * | A set of variables in requested data |
| | Purpose of data use | Intended use or purpose of requested data |
| Providable data | Data name * | The name of providable data |
| | Variables * | A set of variables in the providable data |
| | Data outline | Detailed information on the providable data |
| | Types | The types of data (e.g., text, number, table) |
| | Formats | The formats of data (e.g., CSV, PDF, JSON) |
| | Sharing conditions | The conditions for data providers to exchange data with, or provide data to, other parties |

To describe the information on providable data, we use a part of the data jacket (DJ) [35]. A DJ is a technique for sharing information on data without exposing the actual data, by describing a summary of the data in natural language. The idea of a DJ is to share a "summary of the data" as metadata while reducing the data management cost and privacy risks. In exchanging data summaries, a DJ makes it possible to understand who possesses what kind of data and where the data relate to before actual data exchange. This also allows for an examination of how data are utilized across different fields and organizations. We use the following six description items shown in Table 1 for providable data: "data name", "variables", "data outline", "types", "formats", and "sharing conditions." To identify the data, understand the data structure, and calculate the matching possibility with data requests, "data name" and "variables" are mandatory in providable data and data requests.

## 2.2. A Platform to Match Data Requests and Providable Data

TEEDA is a Web-based application that runs on a Web browser, and input data information is reflected on other users' browsers in real time. A process to provide information on data and a visualization of the TEEDA interface are shown in Figure 1.

We used the description items of variables to calculate the matching between data requests and providable data. For example, if two sets of data information share the common variables of "amount paid", "day", "month", and "purchased item", a link is established between them. This is based on the assumption that the commonality between variable sets indicates similarity between the structural characteristics of the data [25,34]. The studies using ontology-based data access have defined schemas for heterogeneous data integration [36–38]. When discussing data matching through variables, it is debatable whether to use well-defined schemas or natural language concepts. Because the data marketplace is an emerging market, the types of data that are providable to users in the market remain unclear. To allow diverse data with a variety of variables, the data matching in TEEDA does not use schemas; instead, it uses the commonality between variables described in terms of natural language. To express this model, the network of data requests and providable data is represented by an undirected graph, $G := (D_R, D_P, E)$, where $D_R$ represents the set of data requests, $D_P$ is the providable data, and $E$ is the set of edges ($E = \{e_{ij}\}$). An edge is established when the same variables appear simultaneously in a pair of data information ($d_i, d_j \in D$), represented by $e_{ij} = \{d_i, d_j\}$ iff $d_i, d_j \in D$, $d_i \neq d_j$. In the application of TEEDA, to encourage data users and providers to understand the relationships between their own data information and others', links are also established between data requests and between providable data.

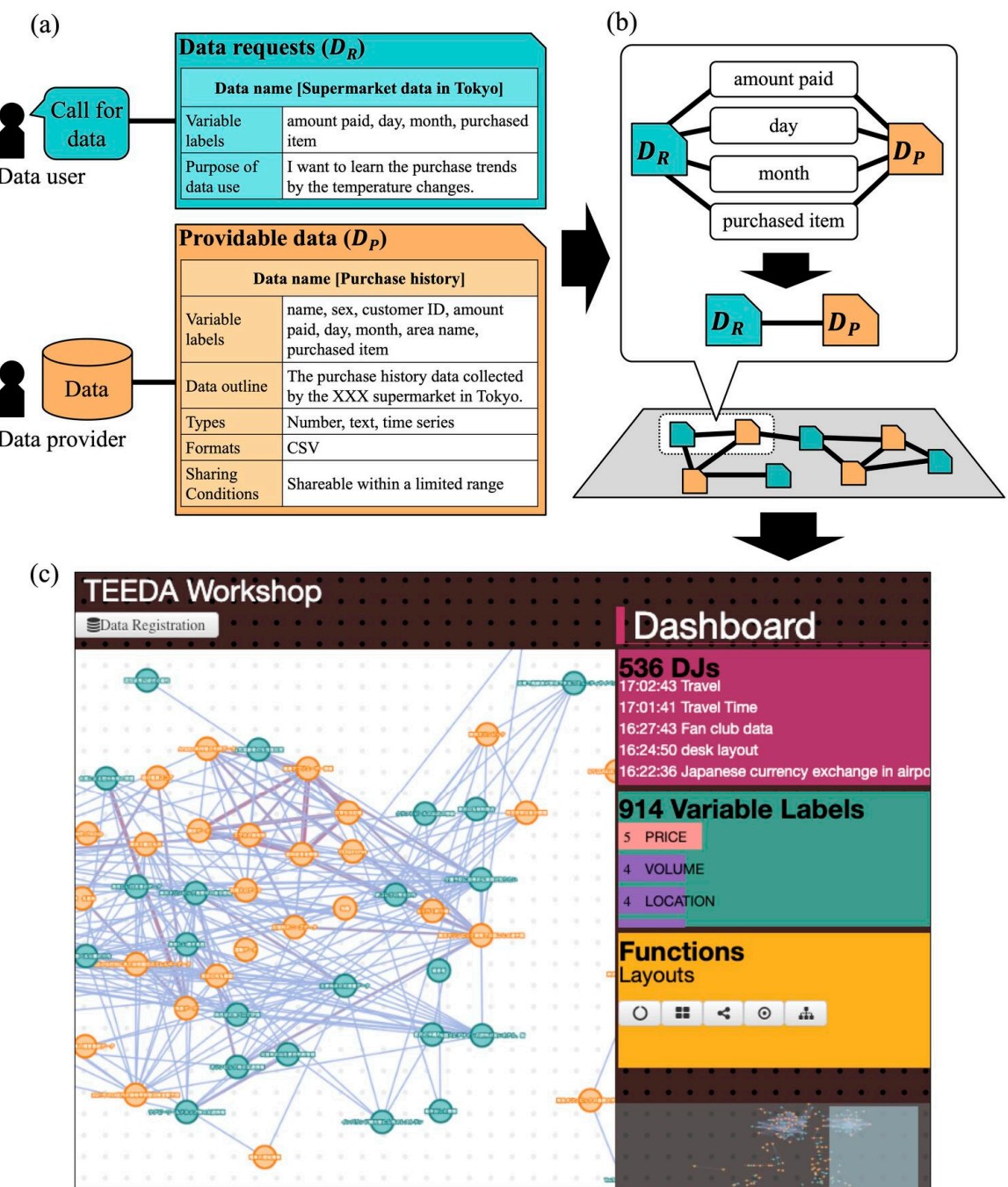

**Figure 1.** The process of (**a**) collecting and (**b**) matching data requests and providable data, and (**c**) the TEEDA interface.

The basic procedure of sharing data information on TEEDA is as follows: (a) data providers can input the information on their data as providable data without sharing the actual data, and data users can call for data as data requests on the same platform; (b) the system calculates the relationships among data information through variables; and (c) the system visualizes the matching network of data information on the interface. To understand the relationships and the matching of data information, we implemented several other functions. Note that the green nodes represent the data requests, and orange nodes represent the providable data. In the matching visualization, the links with more common variables are given greater weight and visualized with thicker edges. When a node is clicked, TEEDA highlights neighboring nodes and displays detailed information on the data requests and

providable data. Note that the system does not show the sample of data in the current version. On the dashboard and toolbox interface, we can understand the number of input data with data names and variables. It is also possible to adjust the network layout manually by dragging and dropping.

## 3. Experimental Details

The aim of the experiment was to understand the structural characteristics of data requests and providable data and examine their matching possibilities. To test its effectiveness, TEEDA was utilized by businesspeople on real-world business data, and data information was acquired in the form of a workshop. The experiment involved 11 workshops conducted using the TEEDA application. Each workshop had an average of 20 participants, with a total of about 220 participants. Participants were men and women (students and professionals) over the age of 20. Initially, we provided a lecture on how to use TEEDA for approximately 30 min. Subsequently, participants input the information on the data requests and the providable data on TEEDA for 30 min through discussion with other participants. The participants input two types of data information: challenging data in real business and academic research due to unavailability as data requests, and the data they could provide to the data marketplace as providable data.

## 4. Results and Discussion

### 4.1. Structural Characteristics of Data Requests and Providable Data

From the experiments, we obtained 536 pieces of data information in total (average at each workshop: 48.7; data requests: 248; providable data: 288). The details of the data information are shown in Table 2. However, the number and type of variables were greater for the providable data, and the variables in data requests seemed to be biased toward certain variables. To understand the difference, we compared the types and the distribution of the variables in detail.

**Table 2.** Characteristic values of data requests and providable data with variables.

|  | Data Request | Providable Data |
|---|:---:|:---:|
| No. of data items | 248 | 288 |
| No. of variables | 1181 | 1606 |
| Types of variables | 779 | 1081 |
| Maximum no. of variables in data | 16 | 52 |
| Minimum no. of variables in data | 1 | 1 |
| Average no. of variables in data | 4.76 | 5.58 |

The frequently appearing variables are shown in Figure 2, and the distributions of the frequencies of variables are shown in Figure 3. We used a rank–frequency plot, which is equivalent to the complementary cumulative distribution function, that is, $\wp(m) \equiv \int_k^\infty p(m')dm' \propto m^{-(\gamma-1)}$, where $\gamma$ is a power-law index [39]. We found that all data information, data requests, and providable data showed a power distribution. In other words, most variables had a low frequency of appearance in both the data requests and the providable data. The variables in the data requests and providable data were not biased toward the specific variables of certain data but had variety. This result suggests that the populations of both data requests and providable data are not composed of data with many high-frequency variables but a small number of diverse variables.

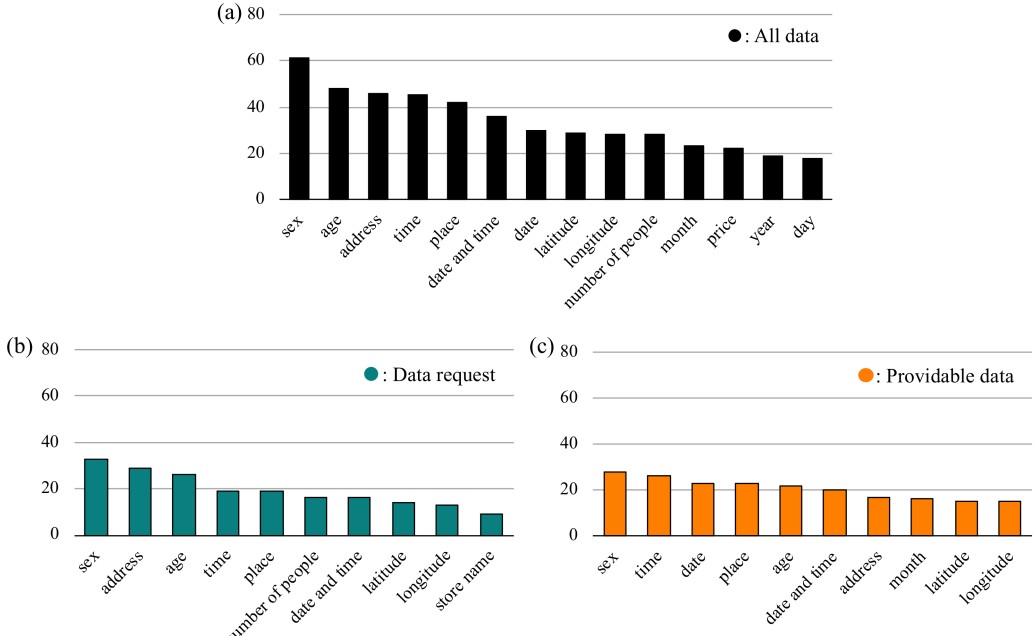

**Figure 2.** (**a**) The top 15 variables in the data requests and providable data. (**b**) The top 10 variables in the data requests. (**c**) The top 10 variables in the providable data.

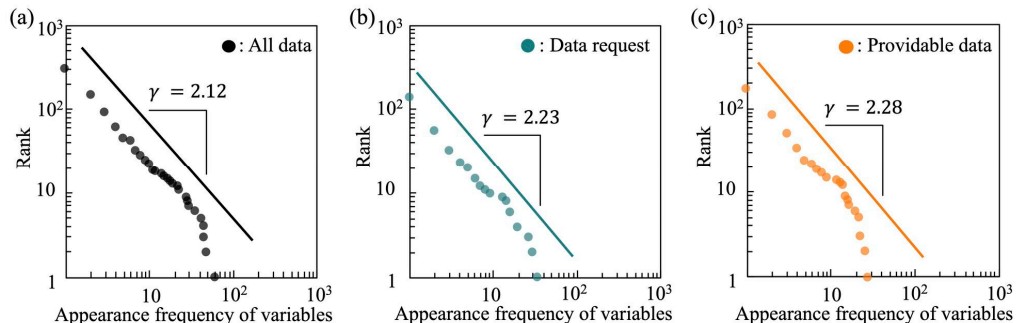

**Figure 3.** Distributions of the frequency of variables in (**a**) all data, (**b**) data requests, and (**c**) providable data.

However, the power-law index of the data requests was smaller than that of the providable data, which appeared in the slope of the distribution. It shows that the ratio of the low-frequency variables is smaller and that the high-frequency variables appeared more in data requests than in providable data. This result suggests that the data requests are composed of more specific variables than the providable data. Further, in Figure 2b,c, it seems that the variables in data requests are biased toward high-frequency variables. The data that are highly required are those including "sex", "address", or "age." The ratio of these variables occupying all data is 7.5% in the data requests and 4.2% in the providable data, which suggests that the variables related to personal data are particularly required.

### 4.2. Distributions and Matching Possibility

Next, we compared the number of variables in each piece of data information. Figure 4 is a log–log graph of the rank–frequency plots of the number of variables in the data requests and providable data. The providable data is in accordance with $\gamma = 3.12$, which is a power distribution. In contrast, the power-law index of the data requests is too high at 4.60, suggesting that it may not be a power distribution. By conducting the Shapiro–Wilk test [40] to verify that the distribution is not the Gaussian distribution, we found that the distribution of the data requests is the Poisson distribution (Figure 4a). The reasons why we applied the Poisson distribution are as follows: (1) the number of variables

contained in the data is a non-negative integer; (2) the lower limit of holding variables is 1 and the upper limit is unknown; and (3) in this observation, the average ($\lambda$) and the variance ($\sigma^2$) were approximately equal ($\lambda = 4.76$; $\sigma^2 = 6.57$) in data requests. On the other hand, the providable data possessed overdispersion ($\lambda = 5.58$; $\sigma^2 = 21.08$), having a very large variance with respect to the average, which is not indicative of the Poisson distribution. In Figure 4a,b, the black dotted lines represent the Poisson distributions given the same amount of data with the same average values of holding variables. In the case of the data requests, the shape is similar. In contrast, although providable data with a small number of variables are similar to the Poisson distribution, they do not follow the line on the data with a large number of variables.

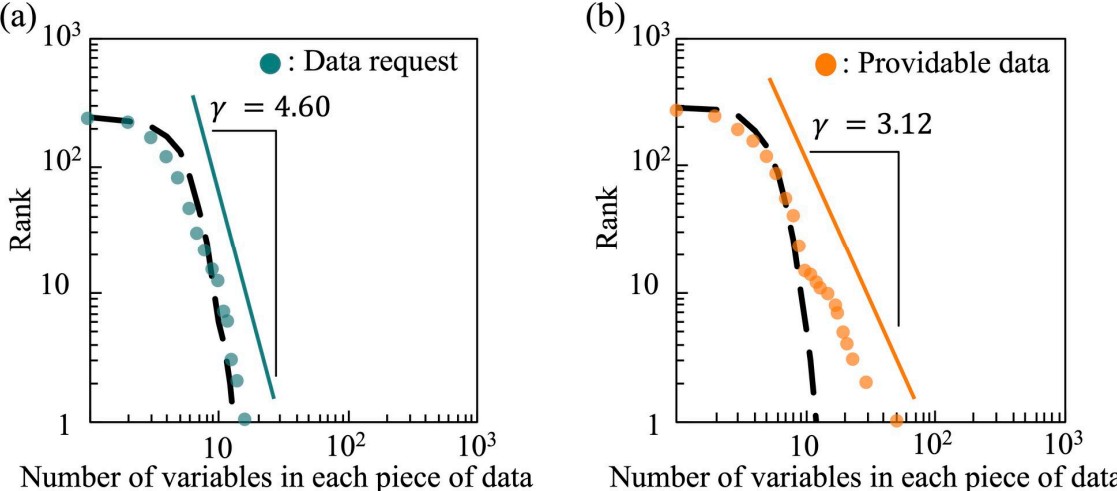

**Figure 4.** Distributions of the number of variables in each piece of data for (**a**) data requests and (**b**) providable data.

Using 1316 providable data, Hayashi and Ohsawa [25] reported that the distribution of the number of variables in providable data is a power distribution. In this study, although the number of providable data is smaller than that in the aforementioned study, we consider that the providable data follow a power distribution from a global perspective. It is important to note that TEEDA is not a platform for data users and providers to input their data information individually, but instead, one where both can refer to the information entered by others interactively on the Web. Although the number of variables in the providable data seems to follow a power distribution globally, the result of our experiment also suggests that the data with a small number of variables were affected by data requests. Comparing the number of links between the data with a small number of variables, i.e., <10 (the part which shows the Poisson distribution in Figure 4b) and those with a large number of variables, i.e., ≥10 (the part which shows the power distribution in Figure 4b), we found that the providable data with a small number of variables matched with 15.2 data requests, while those with a large number of variables matched with 13.3 data requests on average. This result indicates that the data with many variables do not always match with more data requests than those with few variables.

The top five matched pairs by variables are shown in Table 3, where the matching levels are discrete values as the matching was calculated by the co-occurrence of variables. The pairs of data requests and providable data in the table seem to be similar, and the matchings are well established between data information having four or more common variables. It is interesting to note that the top five providable data were the ones with a small number of variables, i.e., <10. On the other hand, when the number of variables is less than two, the matching level is low, e.g., "the bus timetable data" and "the transport history of trucks."

**Table 3.** Top five matched pairs.

| Data Request | Providable Data | No. of Variables in Common |
|---|---|---|
| Human-related data | Human data | 5 |
| Laptop performance data | Laptop performance data | 5 |
| Stock information of convenience stores | Stock information of shops | 5 |
| Health conditions of employees | Human data | 5 |
| Brand image of products | Customers' media contact data | 4 |

Table 4 presents a comparison of the number of links between the data requests and providable data. Interestingly, the number of links between the data requests and providable data is approximately twice that of links among themselves, i.e., between data requests and between providable data. In other words, the commonality of the variables between the data requests and the providable data is higher than that between the data requests themselves and that between the providable data themselves. The small number of links between data requests is due to the variety of users' interests in obtaining data. On the other hand, the smaller number of links between providable data is caused by the variety of variables from different domains. That is, it is suggested that there is a high possibility of potential data matching between data requests and providable data. More interestingly, despite the very large number of links between data requests and providable data, there are only 180 types of variables in common across data requests and providable data. That is, there are only a few types of variables to generate several links between data requests and providable data and match each other strongly.

**Table 4.** Number of links by data combination.

| Type of Data Combination | | No. of Links |
|---|---|---|
| Data request | Data request | 1992 |
| Providable data | Providable data | 2345 |
| Data request | Providable data | 4336 |

On the other hand, the difference in the distribution of the number of variables in each piece of data and few common variables also suggest the possibility of mismatching between data requests and providable data. In the experiment, variables related to personal data such as "sex", "address", or "age" were particularly common among data requests, suggesting that many participants believed personal data are highly required in the market. At the same time, the providable data contained few variables from personal data. Moreover, we found that the number and types of variables were smaller in data requests than in providable data, and the distribution of holding variables was the Poisson distribution. This result suggests that the types and the number of variables that the users want are not as diverse as the providable data. In the current condition of the data marketplace, providable data may have too many variables, and it may be necessary to set a rule to divide and provide data by variables to suit the users' needs.

*4.3. Limitations and Future Work*

In this study, it was suggested that the data requests might follow the Poisson distribution. However, we targeted data and requests under a wide variety of research and business conditions. As various domains use data, it is necessary to observe the distributions by selecting the specific domains, e.g., the manufacturing, disaster prevention, public services, and so on. In addition, the distribution of the providable data seemed to be a power distribution or a mixture of several distributions. Based on these results, two new hypotheses have been proposed.

First, a mixture of several distributions is usually caused by unobserved attributes of data, e.g., the domains of data. For example, the number of variables in data from industry acquired by sensors tends to be diversified. In contrast, data acquired manually, such as the use of a questionnaire to obtain data on social events, have a small number of variables. These mixtures may result in overdispersion.

To avoid this problem, it is necessary to classify providable data by domains using a mixed model, such as the generalized linear mixed model.

Second, it is interesting to note that our result suggested that providable data with a small number of variables matched more data requests than did those with a large number of variables. This might suggest that data providers might have browsed users' data requests and selected the providable data. Conversely, data users might also have browsed the data information from data providers, clarifying the data they want, and inputted their data requests. Thus, in future study, it is necessary to consider the effect of the interaction of data users and providers.

Additionally, in this study, we assumed many-to-many matching among the data requests and providable data. However, in actual matching in the data marketplace, there are cases where one data request is satisfied by a combination of multiple providable data, which is typical of many-to-one matching. In future work, we will use the preference of data requests to discuss more complex matching, including the concept of market design. In the discussion of matching in this paper, we evaluated discrete values based on the number of variables in the pair of data information. For further discussion, it is necessary to use indicators such as Jaccard or dice coefficient that do not depend on the number of variables.

We discussed the potential for data matching via variables in this paper. Additionally, metadata contain information not just limited to variables, but also including contextual information, types, formats, or sharing conditions. Data networks are known to differ depending on variables and contexts [41], and in the future, it will be necessary to discuss data matching based on commonality or similarity using ontology [36–38] and natural language processing [42] to regulate the expression of data requests and offerings in standardized ways. Expectedly, as data markets are places where various stakeholders exchange information across different fields, it is necessary for actual data providers and users to evaluate TEEDA's matching ability.

## 5. Conclusions

In this study, we developed a matching system, TEEDA, which visualizes users' data match-ups to data providers and makes their needs more apparent. Via experiments, we found the different structures and distributions across data requests and providable data. These findings are important for data platformers and data providers in the market. With the development of data catalogs and data exchange platforms, the opportunities for users to discover the data offered by data providers and domain providers have been on the rise. However, there are very few ways for data providers to learn what kind of data is required. A data marketplace cannot be established with only unilateral provision of data from data providers. We believe the diverse exchange of information among all involved stakeholders will accelerate data exchange in the marketplace. The matching application described in this paper will allow matching between data providers and users and can be expected to promote more data-oriented encounters across different fields.

**Author Contributions:** Conceptualization, T.H.; Formal analysis, T.H.; Project administration, Y.O.; Software, T.H.; Writing—original draft, T.H. All authors have read and agreed to the published version of the manuscript.

**Funding:** This study was supported by the MEXT Quantum Leap Flagship Program, grant number JPMXS0118067246, JSPS KAKENHI (JP19H05577 and JP20H02384), and the Artificial Intelligence Research Promotion Foundation.

**Acknowledgments:** We wish to thank Editage (www.editage.jp) for providing English language editing.

**Conflicts of Interest:** The authors declare no conflicts of interest.

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
