# Peer review of "TEEDA: An Interactive Platform for Matching Data Providers and Users in the Data Marketplace"

_information, doi:10.3390/info11040218_

Round 1

Reviewer 1 Report

General comments:

1. can your TEEDA system show a few lines of sample data, with PII anonymized where appropriate, to give prospective data buyers a feel of the data. seeing is believing.

2. How does TEEDA handle acronyms? e.g., gender --> sex

Specific suggestions:

line.146 "form" -> "from"

line.163 table 1 Providable data - Types - you can add e.g., protobuf to allow for binary data

line.168 "as show in Figure 1, respectively"

line.232 "low frequent change" -> "low frequency"

line.293-297 "Interestingly, the number of links between the data requests and providable data is approximately
twice as large as those between data requests and between providable data. In other words, the
commonality of the variables between the data requests and the providable data is higher than that
between the data requests and that between the providable data."

I suggest simplifying the above as it looks funny:

Interestingly, the number of links between data requests and providable data is approximately twice as large as those links among themselves, i.e. among data requests or among providable data.  In other words, the commonality of variables between data requests and providable data is higher than that
between the data requests itself and that between the providable data itself.

Author Response

Title of the paper:

TEEDA: An Interactive Platform for Matching Data Providers and Users in Data Marketplace

Dear Reviewers,

I really appreciate editor’s support and the kind review from the reviewers. According to the reviewers’ comments, we modified the following points. Thank you again for your insightful and valuable comments.

In response to Reviewer 1:

Comment 1:

Can your TEEDA system show a few lines of sample data, with PII anonymized where appropriate, to give prospective data buyers a feel of the data. seeing is believing.

Answer:

Thank you for the insightful comment. In the current version, TEEDA does not have the function to show the sample data. I added the explanation in line 200-201 (pp.6) and would like to consider implementing the function in the next version.

Comment 2:

How does TEEDA handle acronyms? e.g., gender --> sex

Answer:

The acronyms are one of the biggest issues in data matching through variables. As we wrote in pp.4-5 line 173-180, “the data marketplace is an emerging market, the types of data that are providable to users in the market remain unclear. To allow diverse data with a variety of variables, the data matching in TEEDA does not use schemas, and instead, uses the commonality between variables described in terms of natural language.” Also, according to the reviewer’s comment, we added the description to avoid the problem in future work (pp.11 line 360-361).

Comment 3:

Specific suggestions: line.146 "form" -> "from", line.163 table 1 Providable data - Types - you can add e.g., protobuf to allow for binary data, line.168 "as show in Figure 1, respectively", line.232 "low frequent change" -> "low frequency", line.293-297

Answer:

I modified the parts. However, the type of data in Table 1 are just the example, and in this paper, we chose text, number, and table as an example. As reviewer mentioned, there many kinds of types of data, such as protobuf, image, network, and so on.

Comment 4:

Specific suggestions: line.293-297 "Interestingly, the number of links between the data requests and providable data is approximately twice as large as those between data requests and between providable data. In other words, the commonality of the variables between the data requests and the providable data is higher than that between the data requests and that between the providable data." I suggest simplifying the above as it looks funny: “Interestingly, the number of links between data requests and providable data is approximately twice as large as those links among themselves, i.e. among data requests or among providable data.  In other words, the commonality of variables between data requests and providable data is higher than that between the data requests itself and that between the providable data itself.”

Answer:

Thank you for the comment. I modified the parts as the reviewer mentioned (pp.9 line 298-301).

In response to Reviewer 2:

Comment 1:

The only suggestion I have for the authors is to consider whether to evaluate if (or not) adding an ontology-based taxonomy for regulating how to express data requests and offerings in a more standardized way.

Answer:

Thank you for your fruitful comment. According to the reviewer’s comment, we add the description in pp.11 line 362-363.

Once again, thank you very much for your insightful and valuable comments. I hope the modifications make the paper satisfactory to the Journal of Information.

At your disposal for any other modification,

Teruaki Hayashi

Reviewer 2 Report

The manuscript presents an approach to data marketplaces, rooted on DJ descriptive solution aimed at sharing user's call for data.

The paper is well written and presents a lot of elements that are worth of attention.

The state-of-the-art section is up-to-date and adequately addressed.

The proposed approach benefits from a technically sound description and the performed analyses allow the reader to achieve several insights about the results.

The only suggestion I have for the authors is to consider whether to evaluate if (or not) adding an ontology-based taxonomy for regulating how to express data requests and offerings in a more standardized way.

Author Response

(The authors gave the same response as above.)
